

# The identification of gene signature and critical pathway associated with childhood-onset type 2 diabetes

Keren Jia[1], Yingcheng Wu[1], Jingyi Ju[1], Liyang Wang[1], Lili Shi[2], Huiqun Wu[2], Kui Jiang[2] and Jiancheng Dong[1]

[1] Medical School of Nantong University, Nantong, Jiangsu, China
[2] Department of Medical Informatics, Medical School of Nantong University, Nantong, Jiangsu, China

## ABSTRACT

In general, type 2 diabetes (T2D) usually occurs in middle-aged and elderly people. However, the incidence of childhood-onset T2D has increased all across the globe. Therefore, it is very important to determine the molecular and genetic mechanisms of childhood-onset T2D. In this study, the dataset GSE9006 was downloaded from the GEO (Gene Expression Omnibus database); it includes 24 healthy children, 43 children with newly diagnosed Type 1 diabetes (T1D), and 12 children with newly diagnosed T2D. These data were used for differentially expressed genes (DGEs) analysis and weighted co-expression network analysis (WGCNA). We identified 192 up-regulated genes and 329 down-regulated genes by performing DEGs analysis. By performing WGGNA, we found that blue module (539 genes) was highly correlated to cyan module (97 genes). Gene ontology (GO) and pathway enrichment analyses were performed to figure out the functions and related pathways of genes, which were identified in the results of DEGs and WGCNA. Genes with conspicuous logFC and in the high correlated modules were input into GeneMANIA, which is a plugin of Cytoscape application. Thus, we constructed the protein-protein interaction (PPI) network (92 nodes and 254 pairs). Eventually, we analyzed the transcription factors and references related to genes with conspicuous logFC or high-degree genes, which were present in both the modules of WGCNA and PPI network. Current research shows that EGR1 and NAMPT can be used as marker genes for childhood-onset T2D. Gestational diabetes and chronic inflammation are risk factors that lead to the development of childhood-onset T2D.

Corresponding author
Huiqun Wu, wuhuiqun@ntu.edu.cn

## INTRODUCTION

Diabetes mellitus is a chronically progressive, metabolic disease that affects multiple organs over a period of time. In a diabetic patient, blood glucose levels are high as the body either fails to produce insulin or cells are not sensitive enough to insulin. To ensure normal blood sugar levels, most diabetic patients have to take lifelong medications that affect the quality of life. *Ingelfinger & Jarcho, (2017)* reported that the incidence of diabetes has increased phenomenally all over the world. It is frightening to contemplate that the prevalence of diabetes had increased by 30.6% from 2005 to 2015 (*Charlson et al., 2016*). The onset of

diabetes is associated with many factors, including other chronic diseases (*Wang et al., 2017*), social conditions (*Whitworth, Mclean & Smith, 2017*), and even global warming (*Blauw et al., 2017*). In summary, diabetes is a chronic disease of complex etiology and poses a threat to global health.

Based on pathogenesis, diabetes can be mainly classified into three types: gestational diabetes, type 1 diabetes (T1D), and type 2 diabetes (T2D). In general, T1D is known as insulin-dependent diabetes and often occurs in children and adolescents. In patients with T1D, beta cells of islets undergo cell-mediated autoimmune destruction. As a result, the pancreas of T1D patients cannot synthesize and secrete insulin on its own. Furthermore, T2D is known as non-insulin-dependent diabetes mellitus, which is the most common type of diabetes. Approximately 90% of diabetic patients are diagnosed with T2D. It often occurs in patients above the age of 35. In these patients, insulin secretion decreases either due to the lack of insulin receptors or impaired insulin receptors. Gestational diabetes is defined as a condition in which a normal woman develops diabetes during pregnancy. Gestational diabetes usually occurs in the middle or late stages of pregnancy. Some studies monitored women with gestational diabetes for six months after delivery. They found that 38 to 100% of these lactating mothers developed T2DM within six months of delivery (*Kim, Newton & Knopp, 2002*). This suggests that there is an intrinsic link between gestational diabetes and T2D. In addition to these three types, there are some rare types of diabetes, including maturity onset diabetes of the young (MODY), maternally inherited diabetes and steroid diabetes. MODY and maternally inherited diabetes are primary and hereditary diseases. The genetic patterns of them are autosomal dominant and maternal inheritance, respectively. Steroid diabetes is a secondary metabolic disorder caused by excessive glucocorticoids in the body.

With the development of precision medicine, researchers can now pay more attention to patients with childhood-onset T2D. *Wu et al. (2017)* noted that although T2D is common in people above the age of 35, the incidence of T2D has continued to rise in children. Both children and adolescents have physiological conditions that are different from those of adults; therefore, researchers must focus on the prevention, clinical diagnosis, and drug treatment of childhood-onset T2D. Researchers in Mexico have conducted several cross-sectional studies on children and adolescents in the area. The results indicate that T2D and metabolic syndrome-related traits were highly inherited in Mexican children and adolescents (*Mirandalora et al., 2017*). *Joyce et al. (2017)* found that statins increased the risk of childhood-onset T2D without causing dyslipidemia. *Lee et al. (2018)* found that the risk of developing T2D increased significantly in children and adolescents with mental disorders when they were exposed to atypical antipsychotics. *Akhlaghi et al. (2016)* analyzed whether the published anti-hyperglycemic drugs were safe and effective for children and adolescents. These studies have partially revealed some links of T2D in children and adolescents. Nevertheless, scientists still do not know the molecular mechanism of childhood-onset T2D. Therefore, it is necessary to identify the genes and pathways related to the pathogenesis of childhood-onset T2D. Then, we can develop special measures for the prevention and treatment of childhood-onset T2D.

**Table 1 The characteristics of included cases and controls.**

|  | Children with T2D ($n = 12$) | Healthy control ($n = 24$) |
| --- | --- | --- |
| Age (year, mean $\pm$ SD) | $14.0 \pm 2.3$ | $11.3 \pm 4.6$ |
| Sex (%female) | 58 | 58 |
| Race | 2 Caucasian | 11 Caucasian |
|  | 2 Hispanic | 7 Hispanic |
|  | 7 African–American | 6 Mixed or unknown ethnicities |
|  | 1 Asian |  |
| BMI (mean $Z$ score $\pm$ SD) | $2.33 \pm 0.32$ | Unknown |
| Initial pH less than 7.3 | 17% | Unknown |
| Initial HbA1c (mean $\pm$ SD ) | $12.2 \pm 1.5$ | Unknown |

**Notes.**

BMI, Body mass index

The microarray data (GSE9006) was used to compare the intrinsic characteristics of following three groups: healthy vs. T1D groups and the T1D vs. T2D groups. Unfortunately, childhood-onset T2D did not attract the attention of researchers. Hence, we still do not know the differences and similarities between healthy individuals and T2D patients. In this study, we identified the differentially expressed genes (DEGs) in the microarray data (GSE9006) of healthy group and T2D group. Then, we searched for genes closely related to T2D by performing weighted co-expression network analysis (WGCNA), which was further used to calculate the co-expression of genes. We performed functional enrichment analysis on DEGs and T2D genes. In addition, we constructed a protein-protein interaction (PPI) network for these genes to identify the ones that showed critical expression. Our research study focused on finding genes or pathways that are closely related to T2D and revealing the underlying molecular mechanisms.

## MATERIAL AND METHODS

### Affymetrix microarray data

*Kaizer et al. (2007)* measured and uploaded the dataset GSE9006 to gene expression omnibus (GEO) (http://www.ncbi.nlm.nih.gov/geo/). This dataset contained the gene expression of peripheral blood mononuclear cells (PBMCs), which were obtained from 24 healthy children, 43 children with newly diagnosed T1D, and 12 children with newly diagnosed T2D. The demographic information of volunteers was mentioned in Table 1. We selected data of healthy children and children with newly diagnosed T2D for further analysis. The dataset was based on the following platform: GPL97 [HG-U133B] Affymetrix Human Genome Array.

### DEG analysis

In this study, we analyzed DEGs in healthy and T2D patients by using the limma package (*Ritchie et al., 2015*), which is the core component of Bioconductor and is widely used to process the microarray data. |logFC| $\geq 0.5$ and $P < 0.01$ was set as the cut-off criteria.

## WGCNA analysis

To determine the co-expression module and the link between the module and the phenotype, we performed WGCNA of the normalized gene expression matrix by using WGCNA package (*Langfelder & Horvath, 2008*). It is important to note that WGCNA package is an effective data mining tool used to identify all kinds of modules that are highly related to a phenotype, including genes, miRNA, and LncRNA. In this study, the genes were divided into 21 modules, including 20 effective modules and one ineffective module (identified as gray module). We calculated the link between each module and co-expressed genes in healthy and T2D groups.

## GO and pathway enrichment analyses

Gene ontology (GO) (*Ashburner et al., 2000*) includes three types of data, namely, cellular components (CC), molecular functions (MF), and biological processes (BP). GO is often used to annotate genes according to a defined set of structured words. The Kyoto Encyclopedia of Genes and Genomes (KEGG) (*Kanehisa & Goto, 2000*) is used to match the information of gene pathways. Reactome is a great tool to obtain gene-related pathways and interactions (*Beloqui et al., 2009*). GO and pathway enrichment analyses are used to help us figure out the information of genes. In this study, GO and pathway enrichment analysis was performed on DEGs and the genes highly related to T2D. $P < 0.05$ was the criterion used for distinguishing non-meaningful pathways. At least two genes were required to enrich each pathway.

## PPI network analysis

GeneMANIA software (*Montojo et al., 2010*) was used to establish gene interactions and to predict gene function. The PPI network was used to analyze DEGs and genes in both the high-associated modules. We set kappa >0.4 and $P < 0.05$ as the criteria. Network analysis was performed by running GeneMANIA software run in Cytoscape application. The critical hubs were the nodes that were highly connected to other nodes.

## Analysis of diseases and transcription factors related to critical genes

The Comparative toxicogenomics database (CTD) (*Davis et al., 2015*) is often used to analyze the link between genes and diseases. In this study, we determined whether the target gene was linked to diabetes. In the CTD database, we searched all the genes that were reportedly associated with T2D and T1D. Then, we matched these genes with our target genes. iRegulon (*Janky et al., 2014*) is a plugin for the Cystoscape application, which uses a set of pre-defined transcription factors (TFs) and direct transcriptional targets to extract information related to TFs in a group of co-expressing genes. Ultimately, the output is a set of transcription tracks and a list of genes associated with the tracks. A network was constructed with the output results. To reflect the reliability of results, the values of relevant parameters were set as follows: the maximum false discovery rate on motif similarity = 0.001; the minimum identity between orthologous genes = 0.05, and the normalized enrichment score (NES) = 5. Further analysis was performed on pairs for which NES $\geq$ 5.

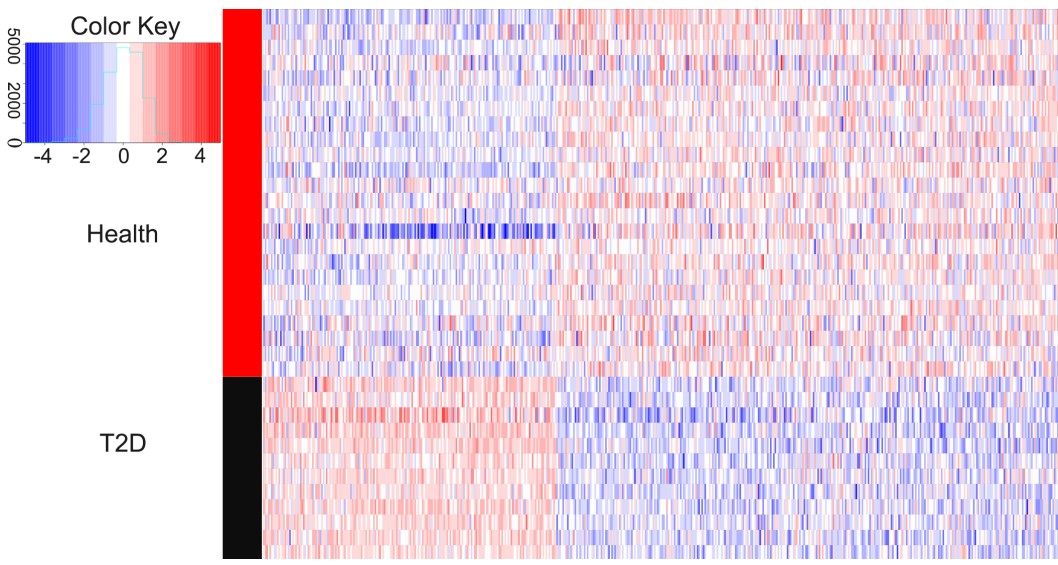

**Figure 1 Heatmaps of differentially expressed genes.** In the right rectangle, red represents a higher expression level, while blue represents a lower expression level. The middle strip indicates the grouping information. The red section is healthy group, while the black section represents the patient group. The right and bottom labels indicate grouping information and gene symbols, respectively. The color key is a histogram. The *X* axis is a numerical value representing the level of gene expression, and the *Y* axis is the number of corresponding squares. This histogram corresponds to the squares.

## RESULTS

### DEG analysis

We compared the gene expression levels of healthy children and children with newly diagnosed T2D. Figure 1 showed that 521 genes had differential expression, including 192 up-regulated genes and 329 down-regulated genes. The average logFC value of up-regulated genes was 0.787, while the average logFC value of down-regulated genes was −0.938. The terms with logFC >1.5 or logFC <−1.5 are EGR1 (logFC = 2.005), HIST1H4PS1 (logFC = 1.793), CELF4 (logFC = 1.665), CXorf56 (logFC = 1.573), PROK2 (logFC = 1.564), ORMDL3 (logFC = −2.496), BF514098 (logFC = −1.846), MRPL39 (logFC = −1.788), LINC00644 (logFC = −1.772), LOC90246 (logFC = -1.770), POLB (logFC = −1.729), AA683356 (logFC = −1.692), MIR497HG (logFC = −1.653), AK026199 (logFC = −1.616), AA885523 (logFC = −1.592), VN1R3 (logFC = −1.588), TOX2 (logFC = −1.579), BF433815 (logFC = −1.576), AW629036 (logFC = −1.555), LRRC45 (logFC = −1.546), LRCH3 (logFC = −1.533), TUG1 (logFC = −1.528), BF110980 (logFC = −1.517), PLD5 (logFC = −1.513) and CCDC102B (logFC = −1.504).

### WGCNA analysis

Under the condition of soft threshold = 6, each gene was divided into 21 modules (20 valid modules and one invalid module) by cluster analysis; the correlation of these genes with T2D phenotype was calculated. Figure 2 shows the correlation results. It can be seen that the blue and cyan modules have a significant correlation with T2D, which are 0.74

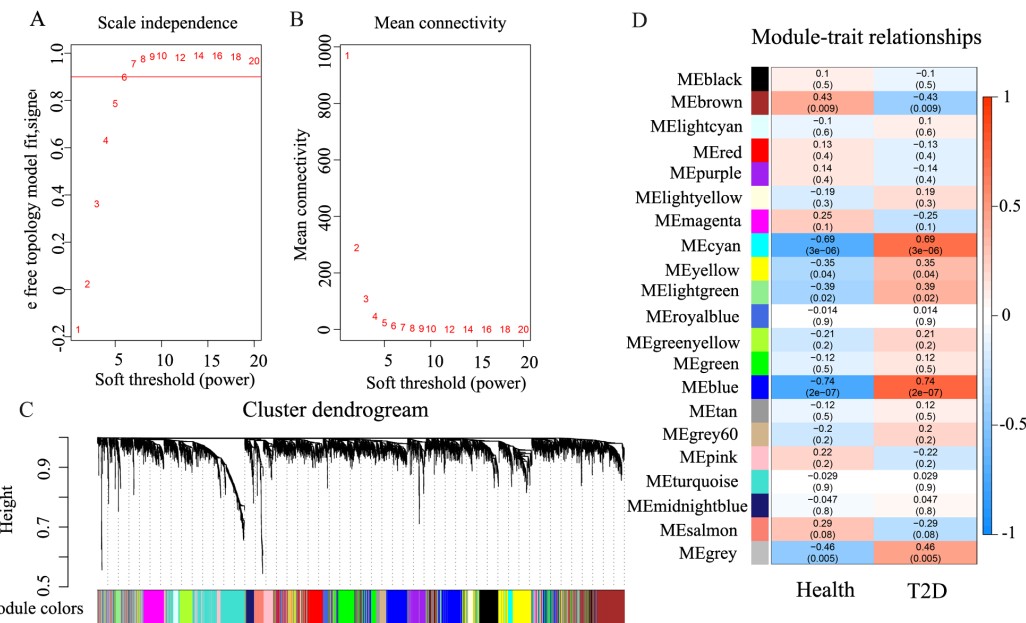

**Figure 2** **The results of weighted co-expression gene network analysis.** An overview of the co-expressed genes in the current study, demonstrating the relevance of gene modules and phenotypes. (A) Scale independence used in WGCNA. (B) Mean network connectivity of soft-thresholding powers used in WGCNA. A soft threshold of 6 is the most suitable value. (C) Cluster dendrogram of the identified co-expression modules. In this figure, each gene is represented as a leaf and corresponds to a color module. Each color indicates that each gene in its corresponding cluster dendrogram belongs to the same module. If some genes have similar changes in expression, then these genes may be functionally related. Moreover, all these genes can further be included into a single module. The grey block represents that the genes that do not co-express with genes of any other color module. (D) Module-trait weighted correlations and corresponding P-values for the identified gene module and their clinical status (healthy and children-onset T2D). The label of color on the left represents the strength of correlation, from 1 (red) to −1 (blue).

($p < 0.01$) and 0.69 ($p < 0.01$), respectively. The two modules contain a total of 636 genes; the blue module contains 539 genes, while the cyan module contains only 97 genes.

## GO and pathway enrichment analyses

GO, Reactome, and KEGG pathway analyses were used to determine the up-regulated genes, the down-regulated genes, and the genes significantly correlated to T2D (blue and cyan). Table 2 displays the GO results, which indicated that the up-regulated genes were highly enriched in cytokine metabolic process, cytokine biosynthetic process, interleukin-1 beta production, positive regulation of cytokine biosynthetic process, and MyD88-dependent toll-like receptor(TLR) signaling pathway. The down-regulated expression of differential genes was mainly associated with the development of lymph vessel and fat pad. In T2D-related modules, the genes either conjugated or removed small proteins to perform functions, such as RNA binding, RNA processing, and protein modification. Some genes were further enriched in organelles' lumen, nuclear region, and other items.

Table 3 presents the results of KEGG and Reactome pathway analysis, indicating the involvement of up-regulated genes in TLR signaling pathway and in diseases of the immune

**Table 2  GO analysis of DEGs and genes in highly correlative module.**

| ID | Description | No. of genes | P-value |
|---|---|---|---|
| **Upregulated genes** | | | |
| GO-BP terms | | | |
| GO:0042107 | cytokine metabolic process | 6 | 3.02E–04 |
| GO:0042089 | cytokine biosynthetic process | 6 | 2.76E–04 |
| GO:0032611 | interleukin-1 beta production | 5 | 2.53E–04 |
| GO:0042108 | positive regulation of cytokine biosynthetic process | 5 | 1.47E–04 |
| GO:0002755 | MyD88-dependent toll-like receptor signaling pathway | 5 | 8.59E–06 |
| GO:0032732 | positive regulation of interleukin-1 production | 4 | 4.47E–04 |
| GO:0045351 | type I interferon biosynthetic process | 4 | 2.19E–06 |
| GO:0032728 | positive regulation of interferon-beta production | 4 | 9.85E–05 |
| GO:0032731 | positive regulation of interleukin-1 beta production | 4 | 2.39E–04 |
| GO:0050702 | interleukin-1 beta secretion | 4 | 4.85E–04 |
| GO:0006491 | N-glycan processing | 3 | 3.84E–04 |
| GO:0034755 | iron ion transmembrane transport | 3 | 4.49E–04 |
| GO:0042228 | interleukin-8 biosynthetic process | 3 | 1.84E–04 |
| GO:0045414 | regulation of interleukin-8 biosynthetic process | 3 | 1.48E–04 |
| GO:0045350 | interferon-beta biosynthetic process | 3 | 3.52E–05 |
| GO:0045357 | regulation of interferon-beta biosynthetic process | 3 | 3.52E–05 |
| GO:0045416 | positive regulation of interleukin-8 biosynthetic process | 3 | 5.00E–05 |
| GO:0045359 | positive regulation of interferon-beta biosynthetic process | 3 | 1.48E–05 |
| GO:0045356 | positive regulation of interferon-alpha biosynthetic process | 2 | 3.43E–04 |
| **Downregulated genes** | | | |
| GO-BP terms | | | |
| GO:0001945 | lymph vessel development | 4 | 1.81E–04 |
| GO:0060613 | fat pad development | 3 | 2.62E–05 |
| Blue-Cyan module | | | |
| GO-BP terms | | | |
| GO:0070647 | protein modification by small protein conjugation or removal | 59.00 | 2.74E–05 |
| GO:0032446 | protein modification by small protein conjugation | 51.00 | 2.49E–05 |

**Table 2** (*continued*)

| ID | Description | No. of genes | *P*-value |
|---|---|---|---|
| GO:0006396 | RNA processing | 51.00 | 5.13E–05 |
| GO:0006397 | mRNA processing | 35.00 | 2.44E–06 |
| GO:0008380 | RNA splicing | 29.00 | 7.10E–05 |
| GO:0007034 | vacuolar transport | 13.00 | 5.86E–05 |
| GO:0007041 | lysosomal transport | 11.00 | 1.04E–04 |
| GO:0007029 | endoplasmic reticulum organization | 8.00 | 3.32E–05 |
| GO:0071786 | endoplasmic reticulum tubular network organization | 5.00 | 9.27E–05 |
| GO-CC terms | | | |
| GO:0043233 | organelle lumen | 216.00 | 1.49E–09 |
| GO:0070013 | intracellular organelle lumen | 216.00 | 1.49E–09 |
| GO:0044428 | nuclear part | 203.00 | 4.75E–13 |
| GO:0031981 | nuclear lumen | 189.00 | 1.36E–12 |
| GO:0005654 | nucleoplasm | 164.00 | 1.55E–11 |
| GO:0044451 | nucleoplasm part | 60.00 | 5.83E–06 |
| GO-MF terms | | | |
| GO:0003723 | RNA binding | 79.00 | 1.65E–05 |
| GO:0019787 | ubiquitin-like protein transferase activity | 31.00 | 2.04E–05 |
| GO:0004842 | ubiquitin-protein transferase activity | 30.00 | 2.01E–05 |

**Notes.**

GO, gene ontology; DEGs, differentially expressed genes; BP, biological process; CC, cellular component; MF, molecular function; cGMP, cyclic guanosine monophosphate.

system. In T2D-related modules, the genes were mainly involved in vasopressin-regulated water reabsorption pathways. The down-regulated expression of differential genes was mainly enriched through chromatin-modifying enzymes, chromatin organization, and mRNA splicing.

## PPI network analysis

We performed PPI network analysis on 112 genes, which were present in both the DEGs and the highly correlated modules. As shown in Fig. 3, we obtained 92 nodes and 254 pairs by setting the PPI score >0.4. An intersection was observed in genes of following types: the genes with logFC >0.6 or logFC <–0.6 in the DEGs, the genes with greater than average degree in the high correlation module, and the genes with greater than average degree in the PPI network. A total of 10 genes were obtained, all of which were up-regulated. The details were presented in Table 4. The 11 genes with the higher degree in the network are C14orf119 (logFC = 16), NAMPT (logFC = 15), NRBF2 (logFC = 14), MTO1 (logFC = 14), PIK3CG (logFC = 13), RNF146 (logFC = 13), VPS50 (logFC = 13), CHM (logFC = 12), GPD2 (logFC = 12), RAB10 (logFC = 12) and ATAD1 (logFC = 12), which have the significant influence on the whole network.

**Table 3  KEGG and Reactome pathway enrichment analyses of DEGs and genes in highly correlative module.**

| ID | Description | No. of genes | P-value |
|---|---|---|---|
| Upregulated genes | | | |
| KEGG | | | |
| KEGG:04620 | Toll-like receptor signaling pathway | 5 | 1.16E–03 |
| Reactome Pathways | | | |
| R-HSA:168898 | Toll-Like Receptors Cascades | 6 | 1.31E–03 |
| R-HSA:8876198 | RAB GEFs exchange GTP for GDP on RABs | 5 | 6.00E–04 |
| R-HSA:5260271 | Diseases of Immune System | 3 | 1.09E–03 |
| R-HSA:5602358 | Diseases associated with the TLR signaling cascade | 3 | 1.09E–03 |
| Downregulated genes | | | |
| Reactome Pathways | | | |
| R-HSA:3247509 | Chromatin modifying enzymes | 10 | 3.10E–04 |
| R-HSA:4839726 | Chromatin organization | 10 | 3.10E–04 |
| R-HSA:72163 | mRNA Splicing—Major Pathway | 8 | 3.76E–04 |
| R-HSA:72172 | mRNA Splicing | 8 | 5.00E–04 |
| Blue-Cyan module | | | |
| KEGG | | | |
| KEGG:04962 | Vasopressin-regulated water Reabsorption | 3 | 1.12E–03 |
| Reactome Pathways | | | |
| R-HSA:909733 | Interferon alpha/beta signaling | 3 | 4.08E–03 |
| R-HSA:8852135 | Protein ubiquitination | 3 | 5.55E–03 |
| R-HSA:8866652 | Synthesis of active ubiquitin: roles of E1 and E2 enzymes | 3 | 3.60E–04 |
| R-HSA:983152 | Transfer of ubiquitin from E1 to E2 | 3 | 1.05E–03 |
| R-HSA:193648 | NRAGE signals death through JNK | 2 | 3.10E–02 |
| R-HSA:933528 | Interaction of MEKK1 with TRAF6 | 2 | 5.91E–04 |
| R-HSA:933530 | Activation of IKK by MEKK1 | 2 | 1.15E–03 |

**Notes.**
KEGG, The Kyoto Encyclopedia of Genes and Genomes; DEGs, differentially expressed genes.

## Analysis of the transcription factors of maker genes

A total of 12 genes were used for further analysis. Among them, 10 important genes were obtained from PPI analysis. The remaining two genes were DEGs with large values of logFC, namely, EGR1 (logFC = 2.00) and NAMPT (logFC = 1.32). In the CTD database, these 12 genes are reportedly linked with T2D. In addition, we analyzed and compared their reports in T1D group. The results were shown in Table 5.

As shown in Fig. 4, we analyzed the transcription factor regulatory network of these genes. We found that NES ≥ 5 was for the following transcription factors: DEAF1 (NES = 7.114, degree = 6), GMEB2 (NES = 6.595, degree = 4), MAF1 (NES = 5.641, degree = 3), NKX3-2 (NES = 5.287, degree = 3), WDR83 (NES = 5.041, degree = 4).

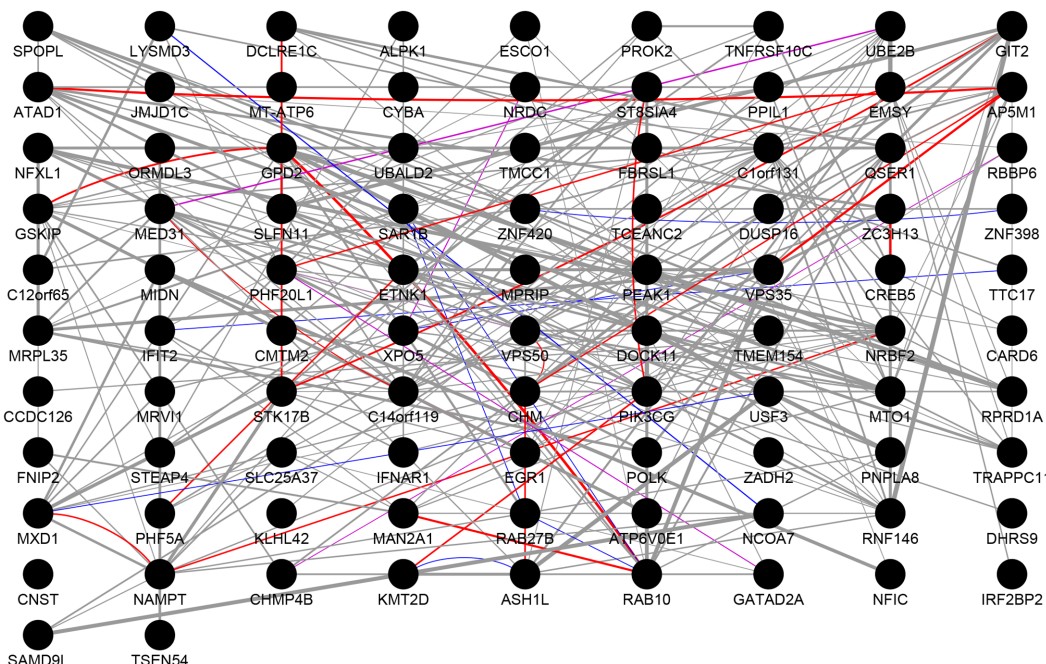

**Figure 3 Protein–protein interaction network.** The results of PPI analysis of DEGs in the highly correlated module. Gray lines, purple lines, red lines, and blue lines represent co-expression, co-localization, physical interactions, and shared protein domains, respectively. The link weight is marked with lines of different thicknesses.

**Table 4 Information of critical genes.**

| Gene symbol | WGCNA module | logFC | WGCNA degree ($n$) | PPI degree ($n$) |
|---|---|---|---|---|
| TRAPPC11 | blue | 1.292140612 | 193 | 8 |
| CHM | blue | 0.801222752 | 138 | 12 |
| C14orf119 | blue | 0.777600335 | 262 | 16 |
| ETNK1 | blue | 0.744750579 | 161 | 11 |
| GSKIP | blue | 0.697264964 | 317 | 11 |
| PIK3CG | blue | 0.688131993 | 178 | 13 |
| ZNF420 | blue | 0.656561519 | 96 | 7 |
| QSER1 | blue | 0.653280142 | 126 | 10 |
| RAB10 | blue | 0.639129192 | 115 | 12 |
| MRPL35 | blue | 0.638260852 | 240 | 7 |

**Notes.**
WGCNA, weighted co-expression network analysis; logFC, $\log_2$ fold change; PPI, protein–protein interaction network; degree, the number of related genes in a given analysis.

**Table 5   No. of references of genes in T2D or T1D.**

| Gene | T2D(n) | T1D(n) |
|---|---|---|
| EGR1 | 104 | 14 |
| NAMPT | 99 | 26 |
| MRPL35 | 27 | 2 |
| QSER1 | 25 | 1 |
| ETNK1 | 11 | 2 |
| RAB10 | 9 | 5 |
| CHM | 6 | 2 |
| PIK3CG | 6 | 6 |
| TRAPPC11 | 4 | 2 |
| C14orf119 | 4 | 1 |
| GSKIP | 1 | 2 |
| ZNF420 | 1 | 1 |

**Notes.**
The number of references related of T2D or T1D is given by the comparative toxicogenomics database.
T1D, type 1 diabetes; T2D, type 2 diabetes.

## DISCUSSION

In humans, early growth response 1 (EGR1) is a protein-coding gene. Previous studies have reported that glucagon can transiently activate EGR1 in liver cells. To mediate glucagon-regulated gluconeogenesis, hepatocytes up-regulate the expression of gluconeogenic genes. By blocking the function of EGR1 gene, we could increase glycogen content in hepatic cells, which would improve the tolerance toward pyruvate and lower fasting blood glucose. The gene EGR1 enhances insulin resistance in T2D patients with chronic hyperinsulinemia. Insulin resistance is one of the most significant causes of T2D (*Shen et al., 2011*). The gene EGR1 promotes the development of gestational diabetes by adversely impacting the glucagon-controlled gluconeogenesis (*Zhao et al., 2016*). In addition, EGR1 promotes also diabetic nephropathy, which is one of the most common complications of diabetes (*Wang et al., 2015*). Several findings have indicated that EGR1 promotes the inflammation within muscle and adipose (*Fan et al., 2013*) and also acts as an inflammatory mediator in reproductive tissues (*Thakali et al., 2014*). Indeed, the expression of EGR1 gene was unregulated in the placenta of obese women. A previous study has shown that the up-regulated expression of IL-8, IL-6, and TNFα was induced by EGR1 gene in BeWo cells (*Saben et al., 2013*). These findings indicated that EGR1 contributed to placental inflammation in obese women. Maternal adiposity must have triggered the expression of EGR1 in umbilical cord tissue (*Thakali et al., 2014*). Current research and previous studies have proved that EGR1 gene has significantly affected many aspects of diabetes and obesity. In multiple tissues of T2D patients, statistically significant differences were found in the expression of EGR1 gene.

Nicotinamide phosphoribosyltransferase (NAMPT) is a protein-coding gene in humans. Previous studies have reported that NAMPT is an adipocytokine that exhibits pro-inflammatory and immunoregulatory properties (*Moschen et al., 2007*) and regulates beta-cell insulin secretion (*Revollo et al., 2007*). Furthermore, NAMPT is involved in insulin

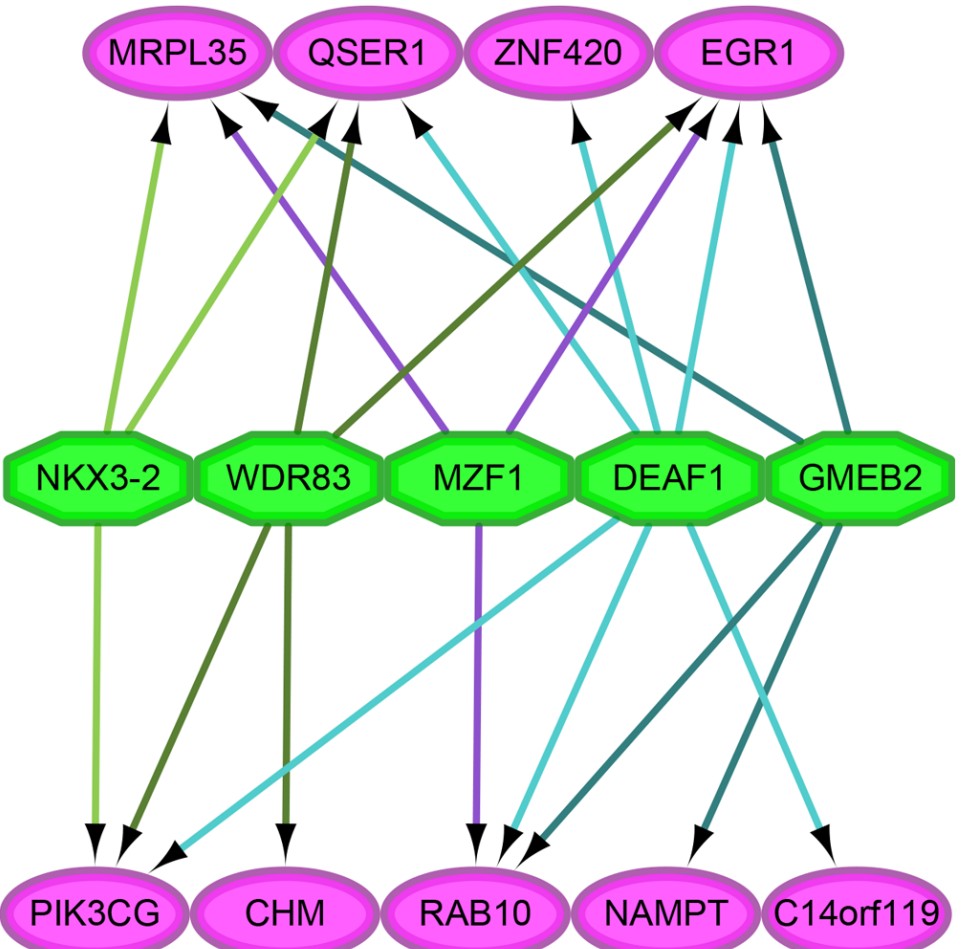

**Figure 4** **The analysis of transcription factor regulatory network.** The pink and green nodes represent the important genes identified by previous analysis and the transcription factors that have regulatory relationships, respectively. Some important genes were not pictured in the network because they were single node, which means that no transcription factor linked to them. The different colored arrows indicate the genes regulated by different transcription factors, making the results easier to observe.

resistance and chronic inflammation, which promotes the development of T2D (*Ma, An & Wang, 2017*; *Jaganathan, Ravindran & Dhanasekaran, 2017*; *Motawi et al., 2014*). Obesity is an important factor that leads to the development of diabetes. In obese and overweight patients with metabolic syndrome, the expression of NAMPT increases in the plasma (*Filippatos et al., 2007*) and simulates the effect of insulin (*Fukuhara et al., 2005*). However, a previous study has reported that NAMPT is related to only T2D and not obesity (*Laudes et al., 2010*). Some studies have reported plasma levels of visfatin, the product of NAMPT, increases in obese people (*Jaleel et al., 2013*), indicating that further research must be conducted to determine the relationship between NAMPT and obesity. Interestingly, the expression of NAMPT was found to be significantly higher in patients with gestational diabetes. This indicates that NAMPT is involved in the molecular mechanism of gestational

diabetes (*Krzyzanowska et al., 2006*). Several studies have reported that Young women with gestational diabetes usually give birth to an overweight baby. This phenomenon has been attributed to intrauterine growth, which increases the prevalence of T2D in the offspring (*Schaefergraf et al., 2005*). Because NAMPT is also conspicuously up-regulated in children with T2D, it is necessary to further study the genetic mechanism of NAMPT in gestational diabetes patients and their offspring. These findings suggest that NAMPT is a marker gene in patients with childhood-onset T2D.

In this study, we also found that chronic inflammation was somewhat related to childhood-onset T2D. Thus, childhood-onset T2D might be caused by multiple risk factors. For the treatment and prevention of childhood-onset T2D, we need to correctly identify the contribution and sequence of risk factors. Hence, further studies must be conducted to confirm whether gestational diabetes and chronic inflammation are closely related to childhood-onset T2D.

According to KEGG, up-regulated genes are mainly enriched in the signaling pathways of TLR. The enrichment results of Reactome pathways indicated that up-regulated genes were related to the TLR Cascades, suggesting that TLRs play a pivotal role in childhood-onset T2D. Reactive oxygen is produced excessively in obese people, which leads to an imbalance in the endogenous antioxidant capacity and causes oxidative stress in adipocytes (*Houstis, Rosen & Lander, 2006*). Obesity also promotes the excessive production of pro-inflammatory adipokines, which further aggravate the chronic inflammation of adipocytes (*Ouchi et al., 2011*). In fact, several studies have explained the association between obesity and chronic inflammation. These studies have further reported that chronic inflammation causes insulin resistance in obese people. Insulin resistance is a precursor to T2D in adults. These studies mainly investigated the adipose tissue of adults. In this study, we found that TLR signaling pathways mainly enriched up-regulated genes in PBMCs. In fact, TLR2 and TLR4 are important cell membrane receptors that elicit innate immune responses to infection (*Tack et al., 2012*; *Chmelar, Chung & Chavakis, 2013*; *Andreas et al., 2003*). Previous studies have shown that TLR2/4 and JNK signaling pathways play a pivotal role in activating CD11c (+) myeloid proinflammatory cells, which further promotes inflammation and subsequent insulin resistance in cells (*Nguyen et al., 2007*). The TLR4 signaling pathway participates in JNK activation and instigates palmitate-induced apoptosis of INS-1β cells. With the knockout of TLR4, we blocked palmitate-induced apoptosis of INS-1 cells; however, no such phenomenon was observed with the knockdown of TLR2 (*Lee et al., 2011*). The inflammatory factors produced during immunization play an important role in obesity-related T2D (*Iiu et al., 2013*). Some immune-related diseases are complications of diabetes. For example, some diabetic patients may develop a chronic airway inflammation, which further causes asthma. The above results indicate that children may develop chronic inflammation, which further induces insulin resistance and promotes the development of T2D. In addition, we also found that several other genes, such as PIK3CG, ZNF420 were also functionally related to inflammation and immune response (*Hawkins & Stephens, 2007*; *Tian et al., 2009*). Therefore, this result further validates our findings.

In previous studies, animal models or elderly patients were investigated to elucidate the mechanism of T2D. Ours is the first study to partially elucidate the molecular mechanism of childhood-onset T2D with the help of bioinformatics. We found the marker genes (EGR1, NAMPT) and TLR signaling pathways of childhood-onset T2D. The up-regulated expression of EGR1 and NAMPT in PBMCs seems to be a gene marker of childhood-onset T2D. The results were compared with the gene expression of middle-aged and older patients with T2D. Thus, there is a genetic mechanism of NAMPT in gestational diabetes patients and their offspring. Such an offspring will have an increased risk of developing childhood-onset T2D. Children are less affected by environmental factors than adults, which allows us to speculate that genetic factors have more influence on childhood-onset T2D. Our findings suggest that NAMPT may be the key to understanding this issue. In addition, TLRs are important proteins involved in non-specific immunity and are a bridge linking specific and non-specific immunity. TLR4 recognizes not only foreign pathogens, but also endogenous substances and degradants. Given the differences in the immune system between children and adults, as well as the special circumstances in which the fetus grows in the uterus, we believe that in children with T2D, the factors that activate the TLR signaling pathway cannot be equated with those of adults. The above results only analyze the underlying mechanisms of EGR1 and NAMPT in children with T2D. We present more details as well, including genes with higher logFC and specific logFC values, genes with higher degrees and associated values, and information on transcription factors, which provide ideas for subsequent research on childhood-onset T2D. It must be emphasized that the physical condition of a child is very different from that of an adult. Therefore, future research studies must focus on elucidating the mechanisms of occurrence, prevention strategies, and treatment of children and adolescents with T2D.

## CONCLUSION

EGR1 and NAMPT can be used as marker genes for childhood-onset T2D, and gestational diabetes and chronic inflammation are risk factors that lead to the development of childhood-onset T2D.

### Funding

This work was supported by the grant of National Key R&D Program of China (2018YFC1314902), National Natural Science Foundation of China (No. 81501559), Excellent Key Teachers in the "Qing Lan Project" of Jiangsu Colleges and Universities 2019, Undergraduate Research and Innovation Plan Project of Nantong University (2018144). The funders had no role in study design, data collection and analysis, decision to publish, or preparation of the manuscript.

### Grant Disclosures

The following grant information was disclosed by the authors:

National Key R&D Program of China: 2018YFC1314902.
National Natural Science Foundation of China: 81501559.
Excellent Key Teachers in the "Qing Lan Project" of Jiangsu Colleges and Universities 2019.
Undergraduate Research and Innovation Plan Project of Nantong University: 2018144.

## Competing Interests

The authors declare there are no competing interests.

## Author Contributions

- Keren Jia conceived and designed the experiments, performed the experiments, analyzed the data, prepared figures and/or tables, authored or reviewed drafts of the paper, approved the final draft.
- Yingcheng Wu performed the experiments, analyzed the data, authored or reviewed drafts of the paper, approved the final draft.
- Jingyi Ju and Liyang Wang performed the experiments, prepared figures and/or tables, approved the final draft.
- Lili Shi, Kui Jiang and Jiancheng Dong contributed reagents/materials/analysis tools, authored or reviewed drafts of the paper, approved the final draft.
- Huiqun Wu conceived and designed the experiments, performed the experiments, analyzed the data, contributed reagents/materials/analysis tools, prepared figures and/or tables, authored or reviewed drafts of the paper, approved the final draft.

## Microarray Data Deposition

The following information was supplied regarding the deposition of microarray data:
Data are available at the Gene Expression Omnibus, accession number GSE9006.

## Data Availability

Raw data and code are available in the Supplemental Files.

## Supplemental Information

Supplemental information for this article can be found online at http://dx.doi.org/10.7717/peerj.6343#supplemental-information.

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
