# Peer review of "The identification of gene signature and critical pathway associated with childhood-onset type 2 diabetes"

_PeerJ, doi:10.7717/peerj.6343_

## Round 0.1 · original submission · Major Revisions

The manuscript got rather mild reviewing comments, however demanding major revision. Please consider all the remarks. And, please, check the Supplemental materials (including English).

Reviewer 1 ·

Basic reporting

'no comment'

Experimental design

'no comment'

Validity of the findings

'no comment'

Additional comments

I was confused a few moments.
1. The manuscript is devoted to the study of the identification of gene signature and critical pathway associated with children-onset type 2 diabetes. But analysis based in the data of 24 healthy and 12 young patients with T2D.
2. It remains unclear how old healthy people are? If they are adults, how likely is the next? Differences in gene expression are caused by physiological age-related features. For example, enrichment up-regulated genes in cytokine metabolic process, cytokine biosynthetic process, interleukin-1 beta production, positive regulation of cytokine biosynthetic process may be caused by large activation of the immune response to a large number of new antigens in childhood. In adults, the immune response is less active.
3. Under the action of insulin, glucose is utilized by cells from the blood. Therefore, the phrase «In a diabetic patient, blood glucose levels are high as the body either fails to produce insulin or does not utilize insulin effectively.» seems to be not entirely successful. Maybe replace it with another? «In a diabetic patient, blood glucose levels are high as the body either fails to produce insulin or cells are not sensitive enough to insulin.»
4. «Based on pathogenesis, diabetes can be classified into three types: gestational diabetes, type 1 diabetes (T1D), and type 2 diabetes (T2D).» It seems that at least it is necessary to mention the monogenic forms of diabetes.
5. It seems that in the discussion it is necessary to explain that visfatin is a product of the NAMPT gene.
My opinion as medical doctor is related only to clinical aspects of this work.
However expert statistician or bioinformatician sould estimate the research.

Reviewer 2 ·

Basic reporting

1. Please clarify: The dataset GSE9006 reports only 43 newly diagnosed T1D subjects but the manuscript reports it as 81 subjects

2. Please change "Islet B cells" to beta cells of islets in line 46

3. T1D group is misspelled as TID in line 76

4. Please clarify: There are two platforms (GPL 96 and GPL 97) reported in the original dataset but the manuscripts mention only one (GPL 97)

Experimental design

No comment

Validity of the findings

1. In table 3, information is provided for only 10 genes (PROK2 is not listed)

2. Result sections should be more descriptive

3. Please adopt or list the demographics of at least healthy and young T2D subjects as the supplemental table 3 and 4 does not have information about age, BMI and Hb1Ac levels

4. Bottom label in figure 1, that is mentioned as gene symbol is not seen in the figure

5. In figure 1, please include the bar that indicates the range of expression level based on color codes

6. Please provide legends for supplementary tables

7. Figure legends are also missing in the word file submitted and it is only available in the pdf file

Additional comments

The manuscript by Keren Jia et.al., is well written and it provides a good rationale for the study. However, the results section can be described in detail with some more clarification on how they are derived. Incorporating some speculations about significance of the findings in the discussion section will add more value to the manuscript. Hence, listed corrections should be considered/made before publication.

---

## Round 0.2 · Minor Revisions

Both reviewers still have some minor remarks. I believe you'd fix it very fast. However I have to return the manuscript for minor revision.

Reviewer 1 ·

Basic reporting

No comment.

Experimental design

No comment.

Validity of the findings

No comment.

Additional comments

I agree with the corrections made with the exception of two points.
It is necessary to correct the error in the lines 34-35: In a diabetic patient, blood glucose levels are high as the body either fails to produce insulin or cells are not Islet B cells (sensitive enough to insulin).
It is necessary to replace “healthy individuals” in to “healthy children” in the line 96.

Reviewer 2 ·

Basic reporting

1. Please change "Islet B cells" to beta cells in line 37 and always refer it as beta cells

2. Yellow text in line 36 and 37 is inaccurate and must be changed

Experimental design

No cooment

Validity of the findings

No comment

---

## Round 0.3 · accepted · Accept

Previous reviewing remarks were really minor, already fixed in the last version. The reviewers have no more comments. I believe this work should be published in current form now, by this year.

Along with the paper acceptance note, let me wish you Merry Christmas and Happy New Year!

#